# A Reconfigurable Pseudohairpin Filter Based on MEMS Switches

**DOI:** 10.3390/s22249644

**Published:** 2022-12-09

**Authors:** Massimo Donelli, Mohammedhusen Manekiya, Girolamo Tagliapietra, Jacopo Iannacci

**Affiliations:** 1Department of Civil Environmental and Mechanical Engineering, University of Trento, 38123 Trento, Italy; 2Center for Sensors and Devices (SD), Fondazione Bruno Kessler (FBK), 38123 Trento, Italy

**Keywords:** hairpin filters, reconfigurable filters, RF-MEMS switches, microwave systems

## Abstract

This work presents a bandpass-reconfigurable planar pseudohairpin filter based on RF-MEMS switches. Hairpin-line structures are preferred to design microstrip filters because this class of filters offers a more compact size, and, in general, hairpin filters do not need ground connections for resonators. In this work, the U-shape resonators are arranged to obtain an interdigit capacitor to improve the coupling between the resonators. RF-MEMS switches modify the lengths of coupled resonators by adding microstrip segments to control the filter bandwidth, moving the center frequency and the return loss. An experimental hairpin tunable filter prototype based on RF-MEMS has been designed, fabricated, numerically and experimentally assessed, and compared concerning its tunability, quality factor, and capability with standard tunable filters based on PIN diodes. In conclusion, the tunable hairpin filter based on RF-MEMS switches offers the best performance in center frequency tuning range, compactness, and power consumption regarding reconfigurable filters based on standard PIN diodes switches. The obtained results are appealing and demonstrate the capabilities and potentialities of RF-MEMS to operate with the new communication standards that work at high microwave frequency bands.

## 1. Introduction

In the last years, there has been a growing evolution in communication systems. The conventional delivery methods, namely, satellite broadcast, radio relays, and cables, have been using analog transmission for many years, and recently, they have been undergoing a revolutionary change to digital broadcasting. Many countries worldwide are already using digital broadcasting by satellite. These revolutions have increased the bandwidth requirements and have led to higher microwave frequency bands and an increased demand for small-size, high-performance, and low-cost microwave devices and systems. Another important characteristic of modern telecommunications systems is their capability to reconfigure themselves to adapt to a new standard. A typical example of reconfigurable systems is the so-called software-defined radio (SDR) [1,2]. Filters play a key role in such a scenario since they are fundamental in almost all communication systems. Conventional filter design methods and circuitry are well documented in the literature [3,4]. However, the new requirements of microwave filters, in terms of efficiency, bandwidth requirements, compactness, power consumption, reconfigurability, and cost, are meeting new exciting challenges in realizing unprecedented demands and new applications [5,6]. In particular, filtering technology is still focusing on reconfiguring itself to different scenarios keeping high-performance requirements, smaller size, and a lower cost [7,8,9]. Due to the low cost and easy fabrication techniques, planar microstrip filters are preferred. Several types of planar microstrip resonator filters [10], such as resonators filters, open loop resonators filters, and stepped impedance resonator filters have been proposed and successfully used for different applications [11,12,13,14]. However, reconfigurability in RF or microwave systems is a top trend, especially in today’s market. For reusability, the same RF front end then dynamically reconfigured its mode of operation according to the application’s demand and standards. In particular, electronically controlled reconfigurable bandpass filters are essential in future multistandard, multifunction wireless communication systems due to their capability to work on different frequency bands and compactness. Hence, reconfigurable bandpass filters can accommodate various bandwidth requirements of different standards, reducing costs and adapting to regulatory changes [15,16,17,18]. Filters characteristics and bandwidth can be modified by changing the inter-resonator couplings. This method is easily implemented in filters based on coupled lines, where the coupling between the elements depends only on the distance and lengths of the overlapping sections. In particular, the length of coupling sections can easily be changed by using electronic switching devices. Because of their compactness [19,20], hairpin bandpass filters are preferred for coupled lines filters. Hairpin bandpass filters are a folded version of a half-wave parallel-coupled filter. In particular, hairpin filters are derived starting from a parallel-coupled filter and folding the ends of resonators back into a “U” shape; this trick significantly reduces the resonator’s size and the resonator’s quality factor Q [21]. Similarly to coupled line filters, hairpin structures can be tuned by varying the length of the resonators by adding a segment of microstrip lines using suitable electronic switches since electronic switches require driving circuitry that can degrade the Q-factor of the filter’s elements. PIN diodes have also been widely used as electronic switches to develop reconfigurable filters. However, they are not the best solution since they require high currents to operate correctly, and the driving circuitry can strongly reduce the quality factor Q. Some attempts to create reconfigurable filters based on PIN diode switches are reported in [22,23]. Other researchers have used varactors to change the filter characteristics [24]. In recent years, many practical engineering applications have used RF-MEMS and obtained outstanding performance [25,26]. In [27,28,29], preliminary attempts to fabricate reconfigurable hairpin filters are reported, while in [30,31,32,33,34,35], different reconfigurable microwave devices are reported. In Table 1, we have compared the hairpin filter technology with our proposed hairpin filter. In the comparison, our proposed work stands out in many parameters such as center frequency, insertion loss (dB), and compact size of filter. It can be noticed in Table 1 that most of the previous hairpin filters which provided a good insertion loss (dB) worked on a higher frequency and had a larger filter size compared to our proposed work. Previous work did not provide filter reconfigurability except the work in [29]. However, the filter in [29] operated on a different center frequency and the dielectric material was quartz. RF-MEMS are very useful for wireless sensors [36] and reconfigurable antennas [37]. RF-MEMS show very good performance in switching speed (usually ranging from fractions of μs [38] to 300 μs [39]), scattering parameters, and power consumption (aside from leakage currents in the order of dozens of μA, electrostatically actuated RF-MEMS switches present a virtually zero power consumption [40]), and they have been demonstrated to work properly up to 90 GHz [41].

In this work, we present the development of a reconfigurable microstrip pseudohairpin interdigitated bandpass filter based on RF-MEMS switches. The proposed filter prototype is compact, simple, and characterized by a very low power consumption. The filter structure is very simple and cheap since no short-circuit connections are required to realize the structure, which is entirely fabricated by planar fabrication techniques. The paper is structured as follows: Section 2 details the mathematical formulation and a description of the basic reconfigurable filter structure, while Section 3 describes the RF-MEMS switches employed in the final prototype. Section 4 reports the performance of a simulated implementation of the proposed filter, equipped with PIN diodes, that leads to a qualitative and quantitative comparison between the two switching mechanisms and a numerical assessment of the prototype of reconfigurable filter based on RF-MEMS. Finally, Section 5 reports the final considerations and some ideas for further work.

## 2. Mathematical Formulation and Filter Design

The hairpin model is one of the most used microstrip filters, and it is obtained by folding half-wavelength microstrip lines into a U shape. This strongly reduces the dimension of resonating elements, but at the same time, the U shape slightly reduces the quality factor Q of the resonator itself [20]. The hairpin filter was obtained by cascading different U-shaped resonators, and it was characterized by a bandpass behavior. Its geometrical parameters are reported in Figure 1a. The first task was estimating the width of U-shape resonators *w* and of the feeding lines $w_{0}$. They could be easily estimated by the well-known Hammerstad formula [47] by imposing a microstrip impedance $Z_{0}=50\;\Omega$ for the feeding lines and the U-shaped resonators. The length *L* and the height *H* of the U-shaped resonators are reported in the following relations (Equation 1) and (Equation 2):(1)$$L=\frac{\left(\frac{\pi }{180}\right)}{\sqrt{\frac{\varepsilon_{r}+1}{2}+\frac{\varepsilon_{r}-1}{2}\left[{\left(1+12\cdot \left(\frac{h}{w}\right)\right)}^{-\frac{1}{2}}+\Gamma \right]}}\times \theta^{\circ }$$
where $\Gamma =0.04\cdot {\left(1-\left(\frac{h}{w}\right)\right)}^{2}$ if $\left(\frac{w}{h}\right)<1$ and Γ=0 if $\left(\frac{w}{h}\right)\geq 1$, and $\varepsilon_{r}$ and *h* are the relative permittivity and the thickness of the dielectric substrate. *w* is the width of the U resonator microstrip. $\theta^{\circ }$ is the so-called sliding factor equal to $10^{\circ }$ [20].
(2)$$H=\frac{\theta^{\circ }}{180^{\circ }}\times \frac{C}{f_{c}\sqrt{\varepsilon_{eff}}}$$
where *C* is the light velocity and $\varepsilon_{eff}$ is the effective dielectric permittivity. The other geometrical parameters *S* and $S_{p}$, which control the coupling, were empirically estimated with a trial and error procedure. Concerning the design of the pseudohairpin structure reported in Figure 1b, the same formulae as those of the previous configuration were considered, together with the guidelines reported in [48]; however, a tuning phase was mandatory for all the geometrical parameters $S_{p},H,L,w,w_{o},g$ and $S_{1}$.

The last two parameters $S_{2}$ and $S_{3}$ were derived from the previous $S_{1}$ and $S_{p}$. We considered a reference bandpass configuration with the following characteristics: a filter order N=2 (only two U-shaped resonators), a central frequency $f_{c}=3.8$ GHz, a fractional bandwidth FB=15%, and a margin factor MF=1.2. Following relations (Equation 1) and (Equation 2), the length *L* and width *H* of the U resonators were L=14.5 mm and H=2 mm, respectively, for a total resonator length of 31 mm, which corresponded to $\frac{\lambda }{2}$ at the central frequency $f_{c}$ for the considered dielectric substrate, RT/duroid 5880 ($\varepsilon_{r}=2.2$, tan(δ)=0.001, and thickness h=0.8 mm). The other geometrical parameters were $w=w_{0}=0.2\;$ mm, g=0.1 mm, $S_{p}=0.35\;$ mm, $S_{1}=S_{3}=0.5\;$ mm, and $S_{2}=0.4\;$ mm.

A parametric analysis was carried out to show the effects of geometrical parameters $S_{p}$ and $S_{1}$. The data were simulated with Advance Design System (ADS) software from Keysight company. The effects of gap $S_{1}$ between the U-shaped resonators on the scattering parameters were analyzed and are reported in Figure 2a,b. In particular, the return loss $S_{11}$ is reported in Figure 2a and the insertion loss $S_{12}$ in Figure 2b. The parameter $S_{1}$ was varied with steps of 0.2 mm, starting from $S_{1}=0.2$ mm up to $S_{1}=1.4$ mm. Figure 3a,b report the scattering parameters versus the gap $S_{p}$ between the U resonators and the feeding microstrip lines. As it can be noticed, while the effects of the coupling gap *g* were negligible, the $S_{p}$ parameter determined the external quality factor (insertion loss) of the filter so that the insertion loss of the device could be further reduced by increasing this gap. The above parametric analysis provided the initial values and limits for optimizing the geometrical parameters obtained with the ADS optimization tools.

The above designed pseudohairpin bandpass filter was then made tunable by varying the length of the U resonators. In particular, this filter could be made tunable in discrete steps by adding sections of microstrip lines to change the length of the resonators. The reference tunable structure is reported in Figure 4, and it can be noticed that two further degrees of freedom were added, the length of the extra segments $d_{s}$, and the gap where electronic switches would be inserted. The idea proposed in this work was to connect the extra segments of microstrip lines with RF-MEMS. In the following, we summarize the geometrical parameters of the switchable structure of Figure 4 obtained thanks to the parametric analysis combined with the optimization tools provided by ADS software: w=0.2 mm, $S_{f}=0.2$ mm, L=15.5 mm, $S_{1}=S_{3}=0.5$ mm, $S_{2}=0.4$ mm, g=0.1 mm, and the gap between segments $g_{s}=0.5$ mm. Without loss of generality, the tunable structure was equipped with three extra segments that guaranteed four different frequency bands. To add further frequency bands, it is sufficient to add further extra segments of the transmission line.

## 3. RF-MEMS Switch

In this section, the RF-MEMS switch enabling the tunability of the proposed pseudohairpin bandpass filter is described with respect to its physical arrangement and electromechanical and electromagnetic features using suitable simulation results and measurements. The following description is intentionally concise since additional and more detailed considerations regarding the present switch can be found in [49]. From a general point of view, the device was a clamped–clamped structure of a series ohmic type driven by an electrostatic actuation mechanism. This meant that a series connection between the two interrupted sections of the RF signal line was enabled by an ohmic metal-to-metal contact (between the movable membrane and the underlying contact pads) when the buried electrode was suitably biased. Encompassed in a coplanar waveguide (CPW) configuration, the device was characterized by overall compactness, with a reduced physical footprint of 0.7 × 1.3 mm^2^ marking the fabricated samples, like the one reported in Figure 5b. The layout displayed in Figure 5a relies on an 8-mask surface micromachining process detailed in [50,51,52], which was developed on a 6-inch quartz or silicon support, covered by a basic layer of silicon oxide. The silicon oxide was deposited onto the different layers to isolate them, and it was etched (and filled with conductive materials) where an electrical connection must be established between adjacent layers.

The first mask consisted of a polysilicon layer composing the central buried electrode (in pink in Figure 5a) and the serpentine-shaped resistors connecting the movable membrane to the ground planes of the CPW, and the ones laying under the anchoring points of the metallic bridge. The former was meant to keep the membrane to a null voltage while preventing the RF signal from flowing toward the ground. The latter were meant to extend the device’s reliability by acting as microheaters and restoring the membrane to the rest position in case of stiction. Stiction is the missed release of the membrane when the voltage of the buried electrode is set to zero. As described in [53], when stiction occurs (because of microweldings or a trapped charge within the oxide), the voltage applied to the two pads implies a flow of current along the resistor, which results in a thermal expansion of the locked membrane and an induced restoring force. The third mask consisted of a multimetal layer based on aluminum composing the buried sections of the RF signal line (in blue in Figure 5a), ending in correspondence of two contact pads laying on the surface of the silicon oxide, which was made by evaporating gold (fifth mask). The electrical connection between the two interrupted sections of the signal line was provided by the actuation of the suspended membrane, which was deployed onto a photoresist sacrificial layer (sixth mask). Two layers of electroplated gold (seventh and eighth masks) comprised the CPW frame and the movable membrane (in brown in Figure 5a), characterized by a simple design: a central transducer with four slender beams leaning against the two anchoring points.

Regarding the electromechanical features of the discussed switch, the following considerations were based on the comparison between the simulated and the measured curves reported in Figure 6. While the simulations were performed in the Ansys Workbench environment, the experimental I–V measurements report the amount of current flowing from input to output ports along the RF signal line against the bias voltage applied to the electrode. As visible in Figure 6, on the one hand, the nominal actuation (pull-in) of the membrane should take place at 21 V, the amount of voltage at which the membrane would reach its maximum displacement (2.5 μm). On the other hand, the reported measure shows that the current along the RF signal line became non-negligible (due to the pulled-in membrane) only for voltage levels greater than 56 V. The significant discrepancy between the simulated and the measured behavior was caused by the residual stress within the gold layers composing the movable bridge. As reported in more specific works [54,55], the residual stress within thin films is the result of different contributions: the juxtaposition of layers with different amounts of internal stress and the thermal treatments performed during the fabrication, with the latter being the most predominant (and fortunately, controllable). For example, during the sacrificial layer removal, the temperatures and the duration of the operation induce stress which may be of a tensile or compressive type, and it may have a uniform or a gradient distribution along the thickness of layers. In the case of structures of clamped–clamped-type, a substantial compressive stress would lead to an out-of-plane deformation (buckling) of the membranes. The amount of residual stress embedded in the movable structures can be determined by measuring the bending of ad hoc test structures on the wafer of fabricated devices utilizing profilometers. The measured deformation is related to a specific amount of residual stress using suitable analytical models [56,57]. In clamped–clamped structures, the presence of a tensile residual stress is not visually evident since the structures usually maintain their planarity, but they are subject to a built-in tension which increases the spring constant of the suspended structure, and thus the amount of voltage required to actuate them. A nearly 100 MPa tensile stress characterized the membranes of this batch of fabricated samples. This led to a general stiffening of the movable bridge and, consequently, to the increased pull-in voltage of 56 V, as visible in Figure 6; by imposing such an initial stress condition to the 3D model [49], it was possible to achieve a simulated behavior with a negligible discrepancy compared to the experimentally measured one.

Regarding the reported switch’s electromagnetic (S-parameter) features, the following considerations were based on comparing the simulation results obtained in the Ansys HFSS environment and actual measurements over a 110 GHz frequency interval. Such a broad observation range had a twofold purpose: firstly, to demonstrate the wideband and remarkable characteristics of the employed devices, and secondly, to underline the excellent agreement between the outcomes of the simulation and the measurement approaches. Concerning the on-wafer measurement setup, the reported S-parameters were characterized on a probe station with the Agilent (now Keysight) N5227A PNA (programmable network analyzer) microwave network analyzer, working up to 67 GHz, with a frequency up-conversion system up to 110 GHz, and ground–signal–ground (GSG) probes (150μm pitch). We delivered 0 dBm and the setup was calibrated with the line–reflect–reflect–match (LRRM) method up to 110 GHz. The LRRM calibration was repeated twice since the measurement was performed in two frequency ranges, from 10 MHz to 67 GHz, and from 67 GHz to 110 GHz. The reason behind the ripples (visible in some of the curves beyond 50 GHz) could be various, from the wear of the probes to nonidealities introduced by cables and/or connectors, or the calibration method itself. However, it is essential to notice the qualitative agreement between the two kinds of reported curves.

Among the different curves reported in Figure 7, the most interesting feature is the isolation (S21) of the whole device in its OPEN state (when the membrane is in rest position), describing the amount of power leaked through the virtually sealed output port. The overall measured isolation was quite linear up to 50 GHz, going from −28 dB at 10 GHz to −15 dB at 40 GHz and reaching −10 dB at 50 GHz. Beyond 50 GHz, the gap between the two isolation curves increased, with measurement one showing ripples from −20 to −10 dB up to 110 GHz. It was also possible to observe a good agreement between the S11 curves (gaps not exceeding 2 dB) up to 50 GHz, beyond which the measurement appeared to be noisier.

Among the curves reported in Figure 8, it is possible to recognize the return loss (S11) curves and the insertion Loss (S21) curves of the device in its CLOSE state (pulled-in membrane). The former describes the amount of back-scattered power at the input port, indicating the impedance matching of the device over the considered frequency span. The latter describes the impairment of the output power introduced by the device itself. In terms of return loss, it can be noticed that the measurement curve was better than −25 dB up to 40 GHz, not exceeding −12 dB up to 110 GHz. Regarding the insertion loss, the measured values were better than −1 dB up to 40 GHz, reaching −3 dB at 50 GHz. Beyond that frequency value, the curve remained above the −10 dB minimum, reached at 85 GHz. As can be observed, in this case, the simulation results provided a better qualitative prediction, showing a general and recognizable agreement with the provided measurements.

In light of the considerations and the experimental evaluations mentioned above, the considered RF-MEMS switch proved to be a compact and reliability-oriented device, still providing wideband and remarkable electrical features that made it a suitable building block for the proposed pseudohairpin bandpass filter.

## 4. Filter Implementations, Comparison, and Assessment

In this section, the pseudohairpin reconfigurable bandpass filter based on RF-MEMS switches is designed, simulated, fabricated, and experimentally assessed. For the sake of comparison, the tunable filter was first equipped with standard PIN diodes and then with RF-MEMS switches, which guaranteed a higher performance and power consumption. As subsequently detailed, RF-MEMS virtually adsorb no current since voltage signals drive them. In contrast, PIN diode switches require currents of dozens of mA to work appropriately, and the requested amount of current strongly limits the battery life in handheld devices. The filter was simulated with commercial software ADS (from Keysight), which permits realistic simulations, including of real devices (such as the MACOM PIN diodes).

### 4.1. Arrangement Based on PIN Diodes

The PIN diodes used in this simulation mimicked the commercial PIN diode from the MS MACOM company. MACOM offers PIN diodes having forward series resistance of about 5.8Ω, and capacitance of 0.018 pF in reverse bias mode at 3.5 GHz. These characteristics lead to an insertion loss of about 0.51 dB in forwarding bias mode and isolation of about 28 dB in reverse bias mode. The accurate MACOM PIN diodes model can be easily derived from the component’s datasheet and directly imported into the ADS simulator in Touchstone format.

In the proposed filter, the substrate RT/duroid 5880 ($\varepsilon_{r}=2.2$ and h=0.787 mm) was employed. The simulator can consider many types of losses: conductor, dielectric, radiation, and magnetic. Conductor losses depend on surface roughness, frequency (skin effect), and conductivity of the top conductive layer of PCB. The schema of a pseudohairpin reconfigurable filter, equipped with PIN diode switches, is reported in Figure 9. As shown from the ADS filter schema of Figure 9, two PIN diodes were employed in parallel to reduce the series resistance offered by the PIN diodes in switch “ON” mode. In the current era of digitization, it is desirable to have tunable filters which can be controlled by TTL signals, CMOS logic, I2C commands, or serialized inputs. PIN diodes control could be done through CMOS logic.

The biasing circuitry consisted of a resistor (10 Ω) to limit the current, a capacitor (50 pF) to filter the noise on the control signal, and a choke (39 nH) to block RF signals. A high-voltage CMOS logic (3.3 V) was applied to the PIN diode to turn it “ON”, and the capacitor added in series with this PIN diode was a part of the circuit. On the other hand, a low-voltage CMOS logic (0 V) set the PIN diode to the “OFF” mode, disconnecting the microstrip segments attached to the U resonators. In this way, one could control the frequency of operation by switching PIN diodes “ON” and “OFF”. By using only three diodes, four different combinations could be obtained, leading to four switchable frequency bands. It is worth noticing that the proposed structure was a proof of concept, and if more frequency bands are required, it is possible to add further microstrip segments.

The simulated scattering parameters obtained for the four different PIN diode configurations are reported in Figure 10. As it can be noticed from the data reported in Figure 10, the obtained results were quite satisfactory: the passband was relatively flat, and the return loss was well below −15 dB in the filter passband. The reconfigurability obtained by the switches was evident and compelling. All combinations and achieved results are summarized in Table 2. The above result was a reference for the MEMS-based implementation of the filter. Concerning the current consumption, the maximum current was adsorbed when all three PIN diode switches on both sides were activated, leading to a total current consumption of about 198 mA, which might be significant for handheld devices. In the following subsections, a qualitative and quantitative comparison between the performance of switches based on PIN diodes and RF-MEMS justifies the substitution of the switching parts. Eventually, a complete prototype is proposed and experimentally assessed.

### 4.2. Comparison

RF-MEMS show excellent performance compared to PIN diode switches. In particular, while the best RF-MEMS switches can be used at very high frequency, up to 90 GHz [41], the best PIN diode switches can operate in the Ka-band.

RF-MEMS switches virtually require no current to work. The already-discussed RF-MEMS switches were activated with voltages of about $V_{d}=60$ V, so to keep the compatibility with CMOS standard driving circuits and most commercial devices (characterized by a 3.3 V power supply), a step-up DC–DC circuit aimed at increasing the voltage from 3.3 V up to 60 V was mandatory. The previously described RF-MEMS switches employed in this work were designed and produced at the Fondazione Bruno Kessler (FBK) laboratories, and they were successfully used in different practical applications [31,32,33,35,36,37].

To provide a more distinct idea regarding the performance of the discussed RF-MEMS switches in the vicinity of the center frequency $f_{c}$, Figure 11a,b report the measured scattering parameters along a narrower frequency range, up to 15 GHz. In particular, Figure 11a,b report the return and the insertion loss of an RF-MEMS switch in OPEN and CLOSE states, respectively. As can be noticed from the data, the RF-MEMS performance, in terms of scattering parameters, was excellent. For the sake of comparison, a single RF-MEMS switch was evaluated at the center frequency $f_{c}$ and compared with a PIN diode switch, and their measured scattering characteristics are listed in Table 3. For evaluation, each switch was connected to its respective driving circuit: a bias network for the PIN diode and a DC–DC step-up converter for the RF-MEMS device. The switches were measured with a VNA and provided a suitable bias-T to prevent damage.

Given the data reported in Table 3, it can be observed that the MEMS-based switch slightly outperformed the PIN diode one; however, it is also worth noticing that the employed RF-MEMS switch kept their good performance up to high frequencies (insertion loss smaller than −1 dB up to 40 GHz), while the considered MACOM PIN diode could not work along such broad intervals. Moreover, the central frequency of $f_{c}=3.5$ GHz characterizing the filter was relatively low to permit comparisons with commercial PIN diode switches. During the preliminary simulation phase, the Touchstone files of the measured scattering parameters of the RF-MEMS switches were integrated into the ADS simulator to obtain realistic simulations, replacing the MACOM PIN diodes and the driving circuitry. The ADS schema of the MEMS-based pseudohairpin reconfigurable filter is reported in Figure 12. The consequent simulation results are discussed in the following subsection in light of the measurements performed on the final prototype.

### 4.3. Experimental Assessment

As proof of concept, the MEMS-based filter prototype is described and assessed in this section. The filter structure was fabricated with a computer numeric control (CNC) machine, considering the same dielectric substrate used in the numerical assessment, namely, RT/duroid 5880 ($\varepsilon_{r}=2.2$ and h=0.787 mm). Then, the filter structure, RF-MEMS switches, and step-up driving circuits were enclosed in a shielded case to prevent interfering signals and connected to the microcontroller. Figure 13 shows a photo of the filter prototype. The DC–DC step-up converter employed to increase the control voltages from 3.3 V up to 60.0 V was a monolithic integrated circuit, namely, the XC9103 from TOREX company.

The adsorbed current, which is necessary to drive the RF-MEMS devices, was less than 100μA; this was a minimal value compared to the current required by the filter equipped with PIN diode switches, which was about 198.0 mA. The filter structure was very compact, and all the components were enclosed within an overall footprint of 40×25 mm${}^{2}$.

The prototype was equipped with two coaxial SubMiniature Version A (SMA) connectors and connected to a VNA in order to measure the return loss $S_{11}$ and the insertion loss $S_{21}$ for all the four possible switches’ configurations. The results are reported in Figure 14, Figure 15, Figure 16 and Figure 17.

The measured data were compared with numerical data obtained by the ADS simulations. The agreement between numerical and experimental data was significant: only a slight frequency shift could be observed, most likely due to the dielectric permittivity tolerance of the substrate.

The filter capabilities to reconfigure itself are reported in Figure 18a,b, which summarize the measured scattering parameters for all the considered configurations. As expected, when all the switches were OFF (Figure 14), the results were almost the same as the ones obtained for the PIN diode switch-based filter, reported in Figure 10. However, the filter based on RF-MEMS switches showed excellent performance for all the other switches’ configurations, although the RF-MEMS devices were demonstrated to slightly outperform the PIN diode characteristics in the vicinity of the central frequency $f_{c}$. In particular, by comparing Figure 10b and Figure 18b, it is possible to notice that the insertion loss of the MEMS-based prototype in the passband region was better than the one of the PIN-diode-based version for every corresponding state of the reconfigurable filter. The advantage of the proposed prototype became more substantial when considering that the one based on PIN diodes was a simulated implementation. As can be seen in Figure 14, Figure 15, Figure 16 and Figure 17, despite the substantial agreement between measured and simulated curves, the measured S21 curves exhibited some local and limited discrepancies compared to the corresponding simulated S21 curves. More specifically, the measured insertion loss showed a maximum improvement of 1.85 dB (111 configuration reported in Figure 17b), and a maximum impairment of 0.4 dB (001 configuration reported in Figure 15b) compared to the simulated insertion loss. Such an interval of discrepancy would reasonably also characterize a possible comparison between the simulated and a fabricated prototype version of the proposed reconfigurable filter based on PIN diodes. For these reasons, a potential comparison between a prototype based on PIN diodes and the proposed one would more markedly highlight the advantages of this realization both in terms of return and insertion loss. In fact, as visible in Table 4, the results in terms of central frequency $f_{c}$, bandwidth BW, and return and insertion loss were generally better than the data reported in Table 2, while in some cases they were comparable, with a deviation less than 1.5%.

The measured current absorption when all the RF-MEMS were activated (configuration 111) was 110.0 μA, a considerably smaller quantity compared to the PIN-diode-based counterpart, which made this device quite suitable to be adopted in handheld devices. In the future, we are aiming to achieve a larger tunability, by increasing the gaps S1, S2, and S3, by changing the location of the MEMS switches with respect to the Sp gap or by changing the location of the RF-MEMS switches.

## 5. Conclusions

A planar microstrip bandpass pseudohairpin filter, based on RF-MEMS switches and characterized by a reconfigurable bandwidth was proposed in this paper. The central frequency and bandwidth variation were achieved by varying the length of the U-shaped resonators by adding segments of microstrip lines through suitable RF-MEMS switches. RF-MEMS switches allowed the device to maintain a high Q-factor, a low insertion loss, and a very low power consumption compared to standard PIN diode switches. A filter prototype was designed, fabricated, and numerically and experimentally assessed. The filter characteristics could be reconfigured in real time thanks to a microcontroller and suitable firmware. The power consumption, Q-factor, central frequency, and bandwidth control outperformed conventional reconfigurable filters in planar arrangements based on PIN diodes or varactors. The obtained results were auspicious and demonstrated that the proposed filter geometry could be applied to higher-order filters with more reconfigurable states and to future implementations operating at very high frequency bands because of the broadband and remarkable electrical features of the employed RF-MEMS components. Future work will aim to achieve a larger tunability by changes in the design of the proposed filter.

## Figures and Tables

**Figure 1 sensors-22-09644-f001:**
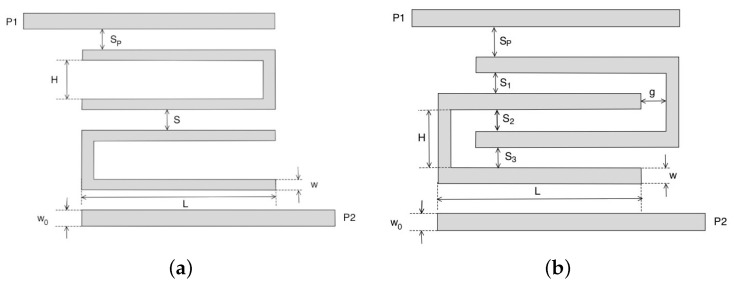
Problem geometries: (**a**) standard hairpin filter, (**b**) pseudo interdigitated hairpin filter.

**Figure 2 sensors-22-09644-f002:**
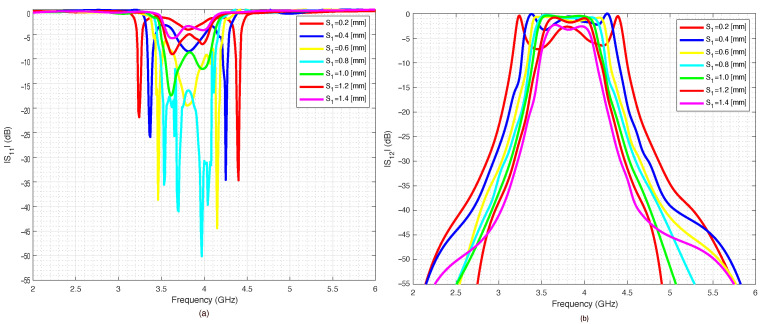
Pseudo hairpin filters scattering parameters versus $S_{1}$ geometrical parameter: (**a**) $S_{11}$, (**b**) $S_{12}$.

**Figure 3 sensors-22-09644-f003:**
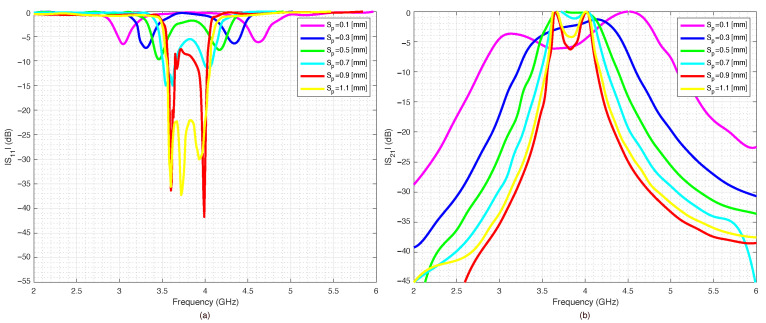
Pseudohairpin filter’s scattering parameters versus $S_{p}$ feeding line distance from resonator: (**a**) $S_{11}$, (**b**) $S_{21}$.

**Figure 4 sensors-22-09644-f004:**
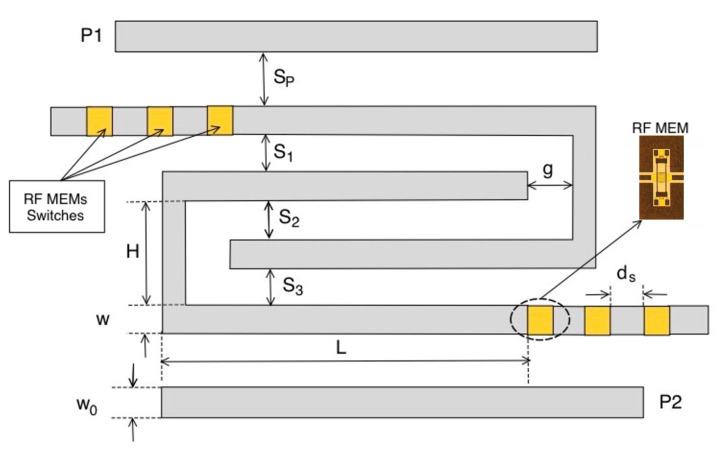
Pseudohairpin interdigitated reconfigurable filter equipped with RF-MEMS switches.

**Figure 5 sensors-22-09644-f005:**
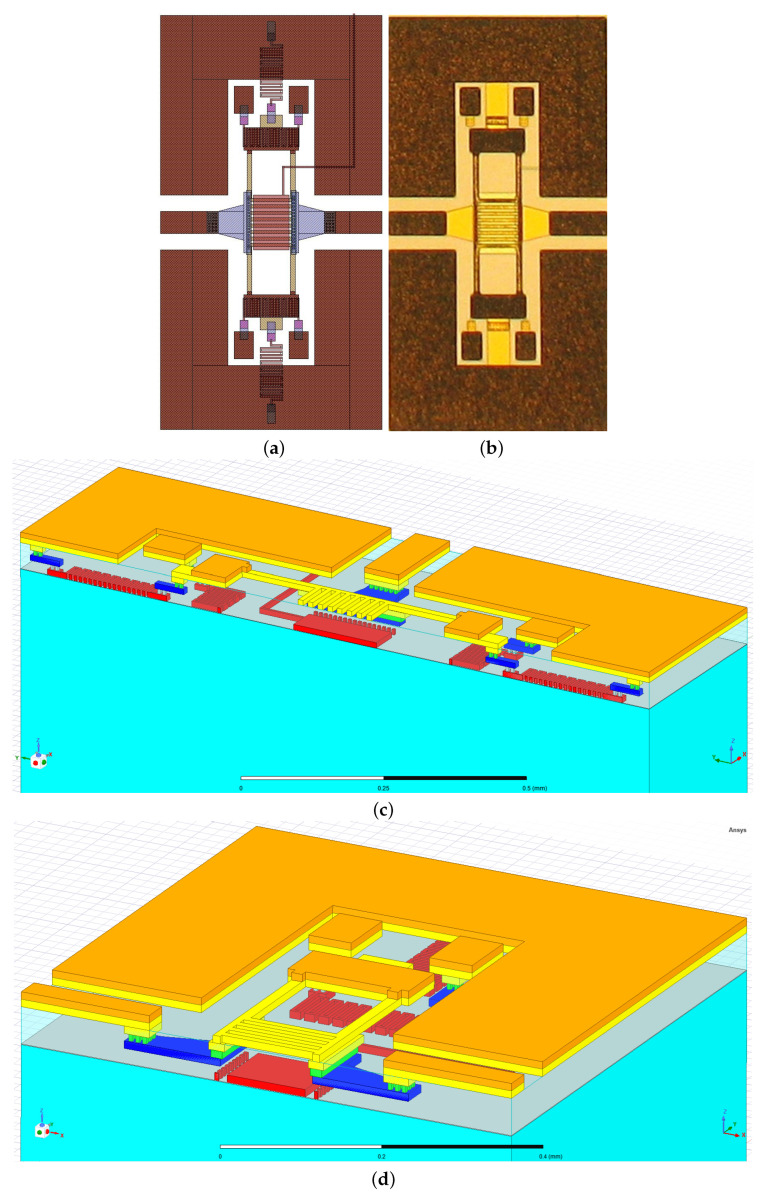
RF-MEMS switch employed in the proposed pseudohairpin bandpass filter: (**a**) complete layout, (**b**) fabricated sample, (**c**) vertical and (**d**) horizontal cross sections of the 3D model in Ansys HFSS environment.

**Figure 6 sensors-22-09644-f006:**
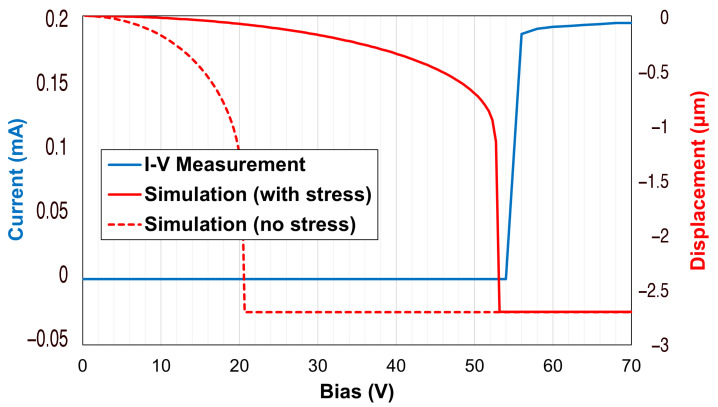
Comparison between the simulated and the measured electromechanical performance of the discussed RF-MEMS switch.

**Figure 7 sensors-22-09644-f007:**
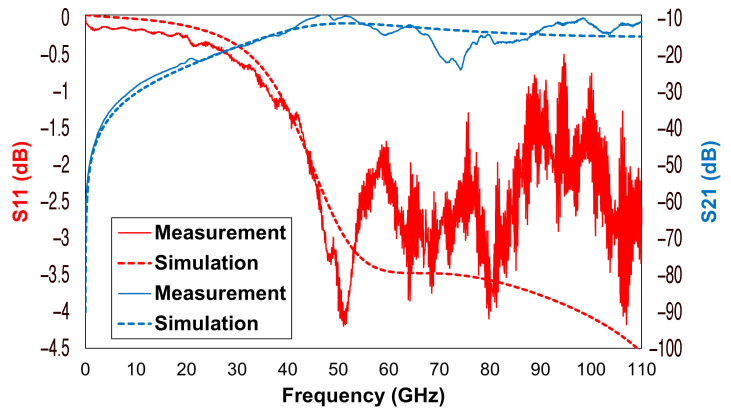
Comparison between the simulated and the measured electromagnetic characteristics of the discussed RF-MEMS switch in its OPEN state.

**Figure 8 sensors-22-09644-f008:**
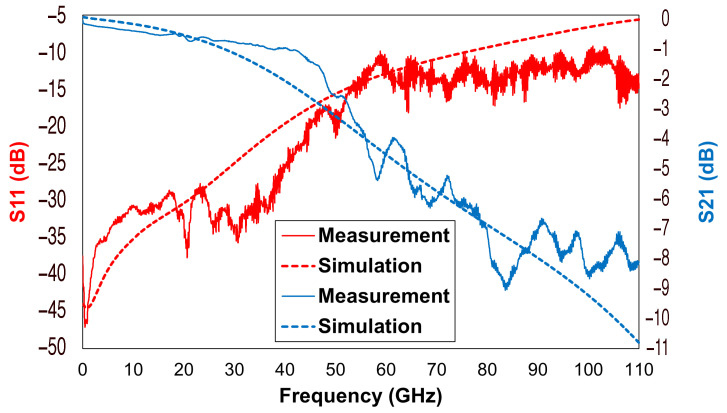
Comparison between the simulated and the measured electromagnetic characteristics of the discussed RF-MEMS switch, in its CLOSE state.

**Figure 9 sensors-22-09644-f009:**
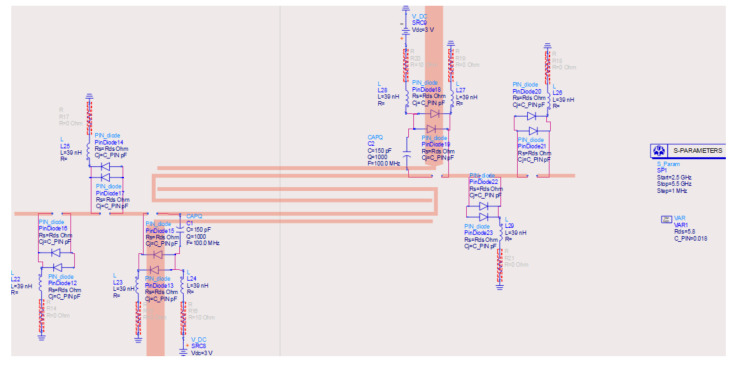
ADS schema of the reconfigurable interdigitated pseudohairpin filter based on PIN diodes.

**Figure 10 sensors-22-09644-f010:**
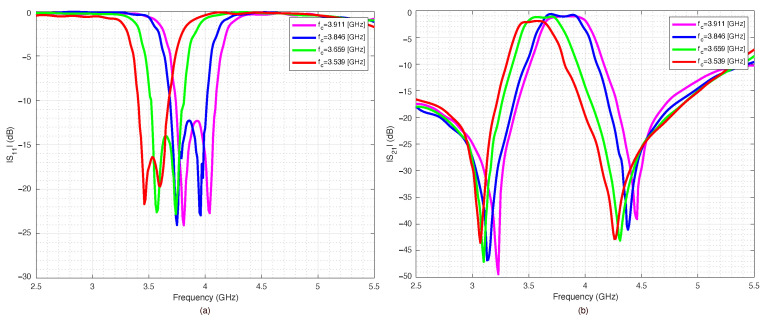
Scattering parameters of the pseudohairpin filter based on PIN diodes: (**a**) return loss $S_{11}$, (**b**) insertion loss $S_{21}$.

**Figure 11 sensors-22-09644-f011:**
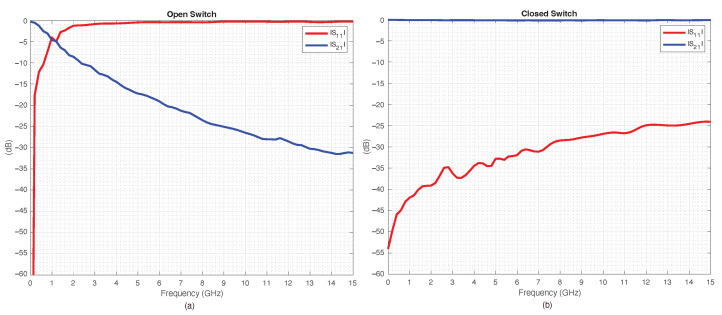
RF-MEMS’ measured scattering parameters: (**a**) OPEN, (**b**) CLOSE states.

**Figure 12 sensors-22-09644-f012:**
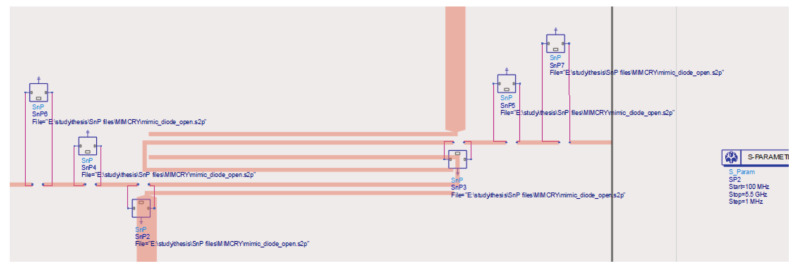
ADS schema of the reconfigurable interdigitated pseudohairpin filter based on RF-MEMS.

**Figure 13 sensors-22-09644-f013:**
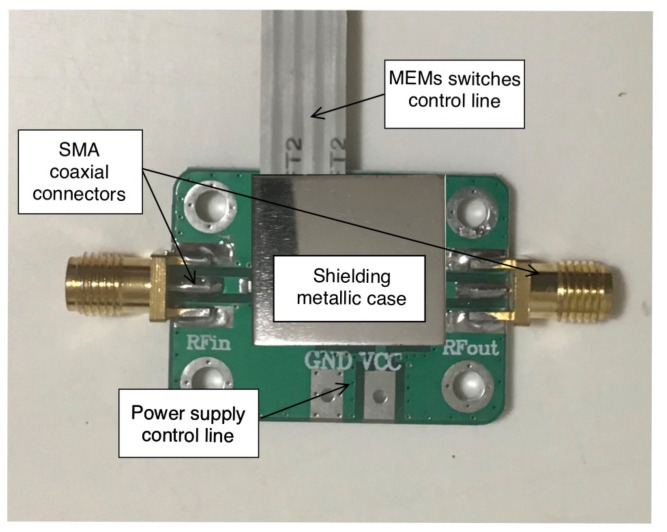
Photo of the reconfigurable interdigitated pseudohairpin filter prototype.

**Figure 14 sensors-22-09644-f014:**
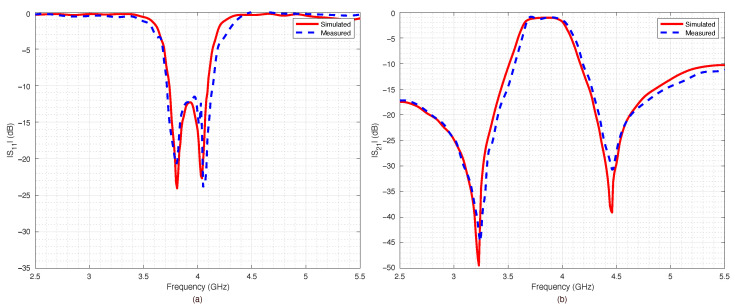
Simulated vs. measured scattering parameters of the pseudohairpin filter prototype: (**a**) return loss $S_{11}$, (**b**) insertion loss $S_{21}$. RF-MEMS switches configuration 000 (all OFF).

**Figure 15 sensors-22-09644-f015:**
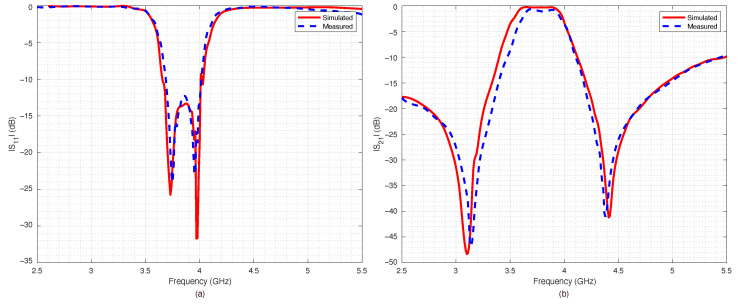
Simulated vs. measured scattering parameters of the pseudohairpin filter prototype: (**a**) return loss $S_{11}$, (**b**) insertion loss $S_{21}$. RF-MEMS switches configuration 001.

**Figure 16 sensors-22-09644-f016:**
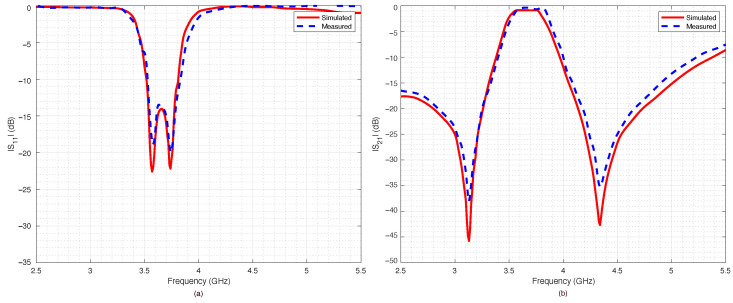
Simulated vs. measured scattering parameters of the pseudohairpin filter prototype: (**a**) return loss $S_{11}$, (**b**) insertion loss $S_{21}$. RF-MEMS switches configuration 011.

**Figure 17 sensors-22-09644-f017:**
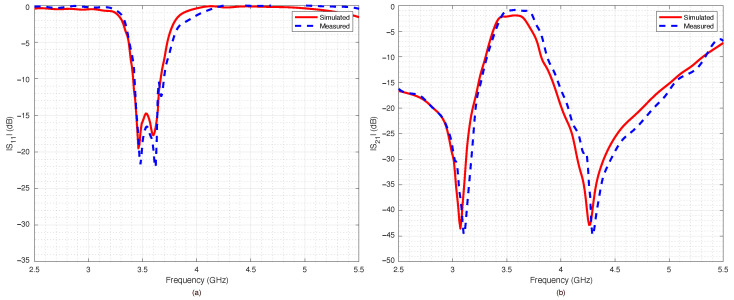
Simulated vs. measured scattering parameters of the pseudohairpin filter prototype: (**a**) return loss $S_{11}$, (**b**) insertion loss $S_{21}$. RF-MEMS switches configuration 111 (all on).

**Figure 18 sensors-22-09644-f018:**
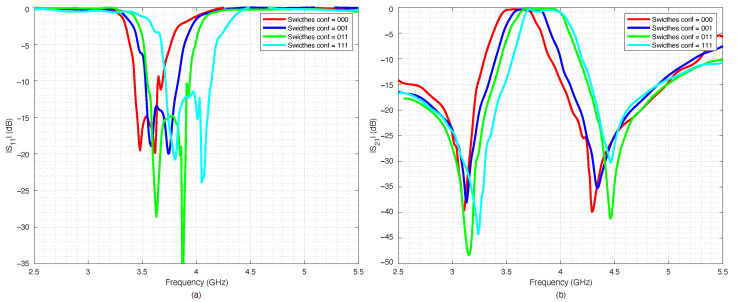
Measured scattering parameters of the pseudohairpin filter prototype for all the considered combinations, (**a**) return loss $S_{11}$, (**b**) insertion loss $S_{21}$.

**Table 1 sensors-22-09644-t001:** Comparison with other hairpin filters.

Ref No	Filter Type	Center Frequency (GHz)	Resonators	Passband Insertion Loss (dB)	Media	Dielectric	Substrate Height
[42]	Hairpin-line, hybrid hairpin-line, half- wave-parallel coupled-line	1.5	4/7	26	Microstrip /MIC	Tellite, $\epsilon_{r}$ = 2.32 Rexohte, $\epsilon_{r}$ = 2.54 99.6% alumina, $\epsilon_{r}$ = 9.7	b = 0.250 in b = 0.250 in h = 0.025 in*10μ
[43]	Quasi-hairpin filter	1.69	2/4	2	Microstrip	RT/duroid 6010, $\epsilon_{r}$ = 10.2	h = 0.635 mm
[44]	Coupled hairpin-line	6.3	1/2	3.2	Microstrip	$\epsilon_{r}$ = 10.8	h = 0.635 mm
[45]	Hairpin-comb	4.32	2	2	Microstrip	$\epsilon_{r}$ = 2.56	h = 0.775 mm
[11]	Notched hairpin	2.14	7	0.69	Microstrip	TLX-8-0620-C2/C2, $\epsilon_{r}$ = 2.55	-
[46]	Hairpin resonator in varactor-tuned	0.3125	4	0.5	Microstrip	RT/duroid 6010LM (Rogers), $\epsilon_{r}$ = 10.2	h = 1.905 mm
[19]	U-shaped hairpin filter	3.1	5	11	Microstrip	FR-4, $\epsilon_{r}$ = 4.4	h = 1.6 mm
[20]	Defected ground structure hairpin filter	9.3	5	3.719	Microstrip	Rogers 5880, $\epsilon_{r}$ = 2.2	h = 1.58 mm
[27]	U-shape RF-MEMS-switch- based hairpin filter	6.2	5	0.3	Microstrip	Silicon, -	-
[29]	Reconfigurable hairpin filter	4.5	2	0.8/1.4	Microstrip	Quartz, $\epsilon_{r}$ = 3.78	h = 0.525 mm
Proposed work	U-Shape RF-MEMS-switch-based pseudohairpin filter	3.8	2	0.8	Microstrip	RT/duroid 5880 $\epsilon_{r}$ = 2.2	h = 0.787 mm

**Table 2 sensors-22-09644-t002:** Summary of simulation results of the tunable pseudohairpin filter equipped with PIN diode switches.

Switch Activation	$f_{c}$ (GHz)	$S_{21}$ (dB)	$S_{11}$ (dB)	BW (MHz)
000	3.911	1.444	12.25	475
001	3.846	1.172	12.68	439
011	3.659	1.308	14.30	403
111	3.539	1.961	16.66	362

**Table 3 sensors-22-09644-t003:** Comparison of scattering parameters between PIN diode and RF-MEMS switches in CLOSE configuration.

Switch	$S_{21}$ (dB)	$S_{11}$ (dB)
PIN	0.40	−15.44
MEMS	0.25	−19.22

**Table 4 sensors-22-09644-t004:** Summary of the measured results of the tunable pseudohairpin filter equipped with RF-MEMS switches.

Switch Activation	$f_{c}$(GHz)	$S_{21}$(dB)	$S_{11}$(dB)	BW(MHz)
000	3.92	1.1	12.2	563
001	3.87	0.25	13.5	465
011	3.65	1.1	14.1	437
111	3.55	2.1	14.8	418

## Data Availability

Not applicable.

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
