# Peer review of "A Reconfigurable Pseudohairpin Filter Based on MEMS Switches"

_sensors, 2022, doi:10.3390/s22249644_

Round 1
Reviewer 1 Report
The present paper is relevant to the design of tunable RF filters for communication applications and provide an interesting comparison between the use of PIN diodes and MEMS switches with the same filter technology. However, previous papers have already implemented similar solutions that combine Hairpin filters with RF MEMS switches. The authors need to explain the differences with previously published works and motivate the novelty of this paper.
Comment 1: In line 67, please, correct switching speed units by replacing “μS” for “μs” and provide values of switching parameters and power consumption based on literature.
Comment 2: In the last paragraph of Introduction, the authors claim that they are presenting “the development of a reconfigurable microstrip pseudo-Hairpin interdigitated bandpass filter based on RF-MEMS switches”. However, there are already many papers that combine a Hairpin EM filter with RF MEMS switches. To understand the novelty of this manuscript, the authors need to compare the proposed filter topology with the ones reported in previous papers and highlight all their differences.
Comment 3: There must be something wrong in Eq. (1) because units do not match between the two sides of the identity. According to the authors, this equation is derived from Hammerstand formula by imposing a microstrip impedance Z=50Ω. In that case, Eq. (1) should depend on frequency and other dimensional parameters from the equivalent capacitor. To make it easier for the reader, I would provide the original microstrip impedance equation and show the entire derivation procedure.
Comment 4: The authors should also provide equations for the rest of introduced variables: w, wo, Sp, S1, and H.
Comment 5: Although the authors refer to [40] in regards to the MEMS switch design, I would add to Fig. 5 two cross sections to better depict the structure from a mechanical point of view: one in the vertical direction that coincides with the electrode and serpentine-shaped resistors, and one in the horizontal direction that coincides with the RF signal line and contact pads. These figures will complement the description of the switch construction in the second and third paragraphs of RF-MEMS switch section.
Comment 6: The authors explain the discrepancies between simulated and measured displacement vs voltage as related to residual stress within the Gold layers composing the movable bridge. This residual stress should generate beam bending, and that magnitude can be easily measured with a profilometer. Have the authors laid out any test structures that could be used to characterize the level of flexural bending and hence extract the actual residual stress? If not, is there any way that authors can determine the stress by directly measuring the switch buckling? I think this validation is important in order to make sure that all the assumptions in the paper are correct.
Comment 7: I would provide the brand and model of VNA used in the measurements. Could those ripples observed in S-parameters beyond 50GHz be due to poor calibration of the RF probe?
Comment 8: The range of tunability shown in Fig. 18 is insufficient for standard communication applications. The filters should be able to tune between adjacent bands with almost no frequency overlap. Can the authors provide a list of recommendations on how to achieve larger tunability by using the proposed pseudo-Hairpin filter topology?
Author Response
[1.0] Thank you for your positive judgment concerning the appropriateness of the proposed methodology, we really appreciate.
In the following, we’ll try to clarify your constructive suggestions aimed at improving the quality of the work:
Comment 1: In line 67, please, correct switching speed units by replacing “μS” for “μs” and provide values of switching parameters and power consumption based on literature.
[1.1] Thank you for pointing this out. The typo at line 67 has been corrected and common values concerning the switching time has been added. As reported in fundamental books concerning RF-MEMS technology, the most frequent actuation mechanism (the electrostatic one) would virtually involve no flow of current between the electrodes, and thus no power consumption. Actually, the negligible flow of leakage current causes a power consumption, which is generally omitted because of its insignificant entity.
Comment 2: In the last paragraph of Introduction, the authors claim that they are presenting “the development of a reconfigurable microstrip pseudo-Hairpin interdigitated bandpass filter based on RF-MEMS switches”. However, there are already many papers that combine a Hairpin EM filter with RF MEMS switches. To understand the novelty of this manuscript, the authors need to compare the proposed filter topology with the ones reported in previous papers and highlight all their differences.
[1.2] Thank you for your valuable comment. We understand that there are different works dealing with reconfigurable hairpin filters, whose geometries are changed or modified by means of electronic switches such as pin diodes or mems switches. However, the proposed filter structure is slightly different, frequency operation is different and more compact with respect to other filters geometries proposed in the scientific literature, moreover the considered mems switches permit to obtain very high performances with respect to the state of the art reconfigurable filters. Thank you.
Comment 3: There must be something wrong in Eq. (1) because units do not match between the two sides of the identity. According to the authors, this equation is derived from Hammerstand formula by imposing a microstrip impedance Z=50Ω. In that case, Eq. (1) should depend on frequency and other dimensional parameters from the equivalent capacitor. To make it easier for the reader, I would provide the original microstrip impedance equation and show the entire derivation procedure.
[1.3] Thank you for your comment. However, the parameters indicated by the reviewers have been numerically estimated with an empirical procedure based on a trial and error procedure without analytical or semi analytical formulas.
Comment 4: The authors should also provide equations for the rest of introduced variables: w, wo, Sp, S1, and H.
[1.4] Thank you. In the paper for w, wo, Sp, S1 and H we followed the guideline presented in reference 45 (mentioned in line 100). To avoid the plagiarism we decided to exclude it from the draft.
- S. Hong and M. J. Lancaster, “Development of new microstrip pseudo-interdigital bandpass filters,” IEEE Microw. Guid. Wave Lett., vol. 5, no. 8, pp. 261–263, Aug. 1995, doi: 10.1109/75.401073.
Comment 5: Although the authors refer to [40] in regards to the MEMS switch design, I would add to Fig. 5 two cross sections to better depict the structure from a mechanical point of view: one in the vertical direction that coincides with the electrode and serpentine-shaped resistors, and one in the horizontal direction that coincides with the RF signal line and contact pads. These figures will complement the description of the switch construction in the second and third paragraphs of RF-MEMS switch section.
[1.5] I agree with you, cross section views of the device would give a clearer insight about the fabrication process and the juxtaposition of the different layers. Unfortunately, I verified that the vertical or horizontal cross section of an object measuring hundreds of μm, does not allow the reader to distinguish layers with a thickness of 630 nm (1 st and 3 rd masks) o 150 nm (5 th mask). In this sense, an image detailing the fabrication steps has been added as the most self-explaining complement to existing images Fig. 5 (a) and (b).
Comment 6: The authors explain the discrepancies between simulated and measured displacement vs voltage as related to residual stress within the Gold layers composing the movable bridge. This residual stress should generate beam bending, and that magnitude can be easily measured with a profilometer. Have the authors laid out any test structures that could be used to characterize the level of flexural bending and hence extract the actual residual stress? If not, is there any way that authors can determine the stress by directly measuring the switch buckling? I think this validation is important in order to make sure that all the assumptions in the paper are correct.
[1.6] You have raised an important point here. Ad hoc test structures are usually embedded on the wafer to measure deformations and to determine the amount of residual stress by analytical models based on the measured deformation. However, such deformation-based modelling of residual stress is not always possible. The lines highlighted in blue colour have been added to better explain the effects of compressive and tensile stress and why such modelling is usually employed just in case of compressive residual stress.
Comment 7: I would provide the brand and model of VNA used in the measurements. Could those ripples observed in S-parameters beyond 50GHz be due to poor calibration of the RF probe?
[1.7] Thank you for this suggestion, The lines highlighted in blue colour have been added to explain the calibration method, the measurement setup, and the possible cause of the visible ripples.
Comment 8: The range of tunability shown in Fig. 18 is insufficient for standard communication applications. The filters should be able to tune between adjacent bands with almost no frequency overlap. Can the authors provide a list of recommendations on how to achieve larger tunability by using the proposed pseudo-Hairpin filter topology?
[1.8] Thank you very much for interesting question. I agree, the filter structure proposed in the paper does not use standard communication. We have used 3.5 GHz frequency or around that range while keeping mind of coming communication standard like 5G (CBRS band). The CBRS band utilizes the 3.5 GHz band. Regarding the frequency overlap, we are working to improve that in the future work. We think there are two ways to achieve larger tunability, 1) By increase the Gap S1,S2, and S3. 2) by changing the location of MEMS switch with respect to Sp Gap. The last and speculated way is by changing the gap between the switch ds.

Reviewer 2 Report
Overall paper is good. Application part of RF MEMS switches in filter is explored and implemented. However, SEM/optical images of the final device with PIN diode and RF MEMS switch without any metallic shield are missing. Is it monolithic or switch and filters are fabricated on different substrate ? If both are fabricated on different substrate, how they are bonded ? and their losses are taken care. Author has claimed that compact tunable filter is made, but comparison with PIN diode dimensions is missing.
Author Response
Thank you very much for your honest judgment concerning the quality of our work. It is worth noticing that, both for PIN diodes and for RF-MEMS, the realization is not monolithic, but we must rely on integration of diverse technologies, with PIN/RF-MEMS mounting on a PCB, or directly on the antenna substrate (realized in any case with PCB-like techniques). No photos of open devices are available. Regarding the PIN diode dimensions, We used a surface mount case devices so the dimensions are quite compact.

Round 2
Reviewer 2 Report
Paper is okay but needs to explain more on packaging and optimization part.
Author Response
Thank you very much for the concerning comment on the optimization part. The filter geometry has been tuned using the Keysight ADS optimization tool and a cost function that considers the scattering parameters at the filter ports. The whole tuning phase required about half an hour on a standard laptop with 16GB RAM.
